# Serological Follow-Up Study Indicates High Seasonal Coronavirus Infection and Reinfection Rates in Early Childhood

Pekka Kolehmainen,[a] Jemna Heroum,[a] Pinja Jalkanen,[a] Moona Huttunen,[a] Laura Toivonen,[b] Varpu Marjomäki,[c] Matti Waris,[a,e] Teemu Smura,[d] Laura Kakkola,[a] Sisko Tauriainen,[a] Ville Peltola,[b] Ilkka Julkunen[a,e]

[a]Institute of Biomedicine, University of Turku, Turku, Finland
[b]Department of Paediatrics and Adolescent Medicine, Turku University Hospital and University of Turku, Turku, Finland
[c]Department of Biological and Environmental Sciences/Nanoscience Center, University of Jyväskylä, Jyväskylä, Finland
[d]Department of Virology, University of Helsinki, Helsinki, Finland
[e]Clinical Microbiology, Turku University Hospital, Turku, Finland

**ABSTRACT** Seasonal human coronaviruses (HCoVs) cause respiratory infections, especially in children. Currently, the knowledge on early childhood seasonal coronavirus infections and the duration of antibody levels following the first infections is limited. Here we analyzed serological follow-up samples to estimate the rate of primary infection and reinfection(s) caused by seasonal coronaviruses in early childhood. Serum specimens were collected from 140 children at ages of 13, 24, and 36 months (1, 2, and 3 years), and IgG antibody levels against recombinant HCoV nucleoproteins (N) were measured by enzyme immunoassay (EIA). Altogether, 84% (118/140) of the children were seropositive for at least one seasonal coronavirus N by the age of 3 years. Cumulative seroprevalences for HCoVs 229E, HKU1, NL63, and OC43 increased by age, and they were 45%, 27%, 70%, and 44%, respectively, at the age of 3 years. Increased antibody levels between yearly samples indicated reinfections by 229E, NL63, and OC43 viruses in 20–48% of previously seropositive children by the age of 3 years. Antibody levels declined 54–73% or 31–77% during the year after seropositivity in children initially seropositive at 1 or 2 years of age, respectively, in case there was no reinfection. The correlation of 229E and NL63, and OC43 and HKU1 EIA results, suggested potential cross-reactivity between the N specific antibodies inside the coronavirus genera. The data shows that seasonal coronavirus infections and reinfections are common in early childhood and the antibody levels decline relatively rapidly.

**IMPORTANCE** The rapid spread of COVID-19 requires better knowledge on the rate of coronavirus infections and coronavirus specific antibody responses in different population groups. In this work we analyzed changes in seasonal human coronavirus specific antibodies in young children participating in a prospective 3-year serological follow-up study. We show that based on seropositivity and changes in serum coronavirus antibody levels, coronavirus infections and reinfections are common in early childhood and the antibodies elicited by the infection decline relatively rapidly. These observations provide further information on the characteristics of humoral immune responses of coronavirus infections in children.

**KEYWORDS** 229E, HKU1, NL63, OC43, seasonal coronavirus, serology, respiratory infection, antibodies, enzyme immunoassay, children

Address correspondence to Pekka Kolehmainen, pekka.j.kolehmainen@utu.fi.
The authors declare no conflict of interest.

The family of coronaviruses consists of enveloped positive-strand RNA viruses, which are further classified into four genera, alpha-, beta-, gamma- and deltacoronaviruses, based on their phylogenetic relationships (1). Human coronaviruses (HCoVs) can cause respiratory infections ranging from subclinical and mild infections to a severe acute respiratory syndrome (SARS) (2). Four of the HCoVs, "seasonal" HCoVs, are low

pathogenic, endemic viruses and their infections are common in all age groups, especially in children under 10 years of age (3). Seasonal HCoVs may be detected in respiratory tract samples from hospitalized children although they are rarely the causative agent of a severe respiratory infection (4). Seroconversion for the seasonal alphacoronaviruses has been estimated to occur on an average at the age of 3.5 years (5), and seroprevalence reaches its maximum at the age of 5 years (6).

Genetically and biologically HCoVs are distinct from each other. All HCoVs can cause respiratory infections (7) although they have differential preferences for host cell tropism and receptor molecules (7, 8). Two of the seasonal HCoV species, 229E and NL63, are members of the genus alphacoronavirus while HKU-1 and OC43 belong to genus betacoronavirus, which also includes Middle East respiratory syndrome virus (MERS), severe acute respiratory syndrome coronavirus (SARS-CoV), and SARS-CoV-2. Obtaining accurate data on the prevalence of seasonal coronavirus infections is challenging since reinfections are common and antibody levels may wane relatively rapidly (9, 10). However, antibody levels remain detectable for up to 2 years (9, 11), which supports the use of serological assays for estimating coronavirus infection burden in early childhood.

This study increases our knowledge on the rate of seasonal HCoV infections in early childhood. We used nucleoprotein-based (N) enzyme immunoassay (EIA) to analyze IgG antibody levels against four seasonal coronaviruses in 140 children in Finland with sequential serum samples collected at the ages of 1, 2, and 3 years in a setting of a prospective follow-up birth cohort.

## RESULTS

**Seasonal HCoV seroprevalence in children.** To study the seropositivity and IgG antibody levels for seasonal HCoVs in young children, we analyzed sequential serum samples collected at the age of 1, 2 and 3 years from 140 children for HCoV N-protein-specific IgG antibodies using EIA. The analysis showed differences in the prevalence of elevated antibody levels between HCoVs (Fig. 1A). In all of the age groups the antibody positivity rate was highest for NL63 (21% in 1-year-old, 31% in 2-year-old, and 57% in 3-year-old children) followed by 229E (16, 27, and 37%), OC43 (14, 25, and 29%), and HKU1 (6, 16, and 14%) (Fig. 1B).

The seropositivity increased with age, except for HKU1, which had a lower seropositivity in 3-year-old children in comparison to 2-year-old children. The cumulative seropositivity at 2 and 3 years of age was higher compared to the yearly seropositivity. This difference resulted from children turning seronegative after the earlier seropositivity. The proportion of children who were seronegative for 229E, HKU1, NL63, and OC43 at 2 years after seropositivity at 1 year was 32% ($n = 7/22$), 63% ($n = 5/8$), 48% ($n = 14/29$), and 47% ($n = 9/19$). Of children who were seropositive for 229E, HKU1, NL63, and OC43 at 2 years, 18% ($n = 7/38$), 64% ($n = 14/22$), 30% (13/44), and 40% (14/35) were seronegative at the age of 3 years. In the 3-year-old children, the cumulative seropositivity was highest for NL63 (70%), followed by 229E (45%), OC43 (44%), and HKU1 (27%).

Altogether, 45% (63/140) of the 1-year-old children had anti-N IgG antibodies against at least one of the seasonal coronaviruses indicating that these children had experienced at least one HCoV infection by the age of 1 year (Fig. 1B). The proportion of children with antibodies for at least one seasonal coronavirus increased to 67% by 2 years and to 84% by 3 years of age. Sixteen percent of the children had no antibodies for any seasonal coronaviruses during the follow-up. Of all 420 serum samples, eight (1.9%) and nine (2.1%) showed IgG antibody levels above the cutoff for MERS and SARS-CoV-2 N proteins, respectively.

**Antibody levels induced by seasonal HCoV infection and reinfection.** To estimate the changes in IgG antibody levels following primary infection, the children were grouped based on the age they were seropositive for the first time. In children who were seropositive for any of the seasonal coronaviruses at the age of 1 year, the geometric mean IgG antibody levels (GMALs) decreased by the age of 2 years (229E 30 to 16 EIA units, $P = 0.029$; HKU1 23 to 10, ns; NL63 33 to 11, $P = 0.004$; OC43 23 to 16, ns; Fig. 2A). In these same children, the GMALs continued to decrease for HKU1 and OC43

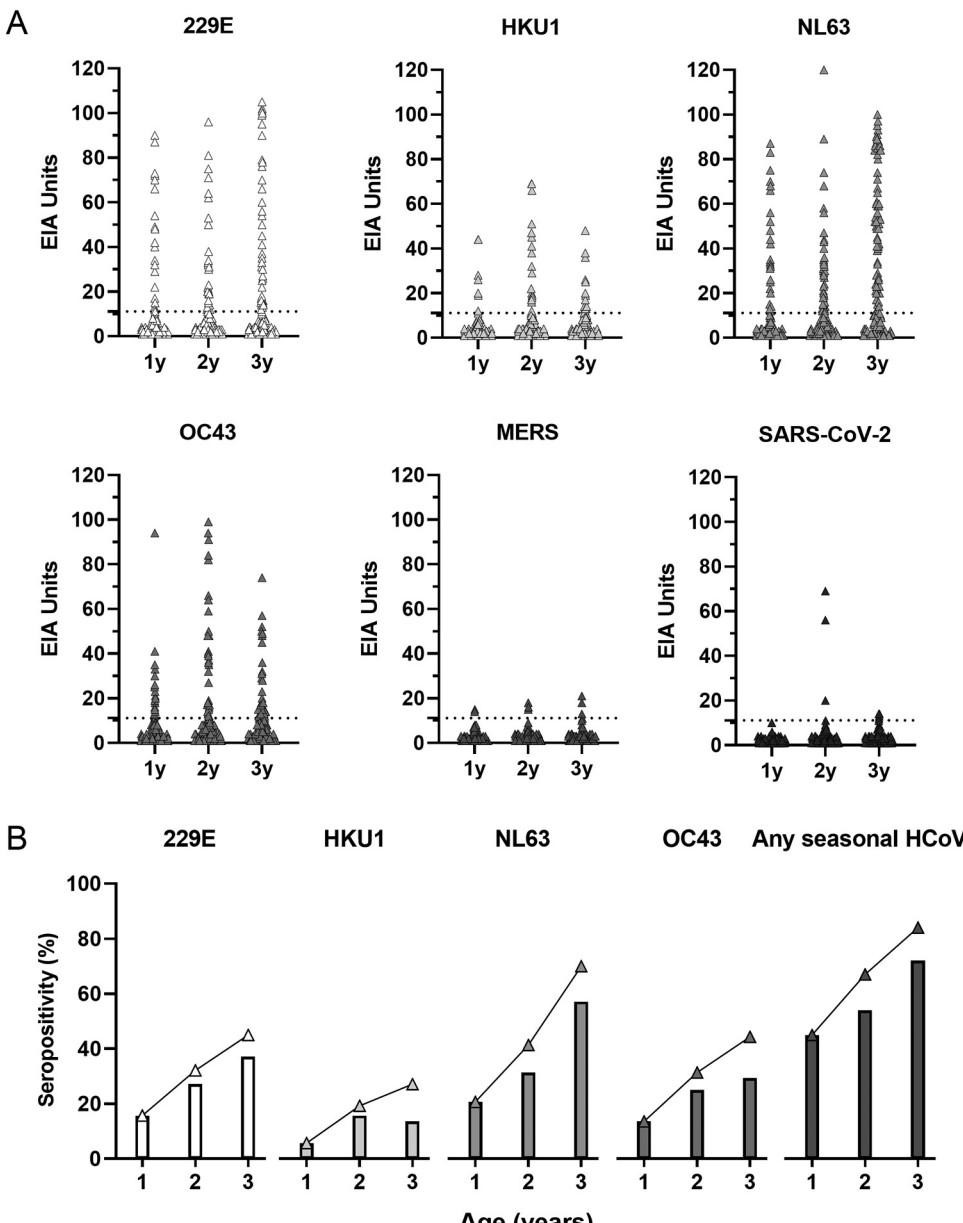

**FIG 1** Seropositivity and IgG antibody responses for six HCoV N proteins in children. Anti-N IgG antibody levels were measured with EIA. Serum samples were collected from 140 children at the age of 1, 2, and 3 years and analyzed for IgG antibodies against 6 HCoV N proteins (A). Cutoff value is indicated with a dashed line. Antibody levels are shown as EIA units, which have been calculated in relation to positive (100 EIA units) and negative (0 EIA units) control specimen pools. Samples with an EIA unit value lower than 1 were given a value 1 enabling the calculation of geometric means. The rates of N IgG seropositivity for seasonal HCoVs, and for any seasonal HCoV in different age groups (shown as bars) and as a cumulative seropositivity (shown as triangles) are shown (B).

from 2 to 3 years (10 to 5 EIA units, ns; 16 to 11, ns, respectively), whereas for 229E and NL63 the antibody levels between individuals varied greatly at 3 years, which resulted in slightly increased GMALs (16 to 20 EIA units, ns; 11 to 22, $P = 0.041$, respectively). Similar trends were observed in children who experienced their primary infection between 1 and 2 years (Fig. 2B): GMALs for HKU1 and OC43 decreased significantly (29 to 7 EIA units, $P < 0.0001$; 32 to 12, $P = 0.008$) from 2 to 3 years, whereas for 229E and NL63 the antibody levels between individuals showed great variation and no signifi-cant changes were observed in GMALs from 2 to 3 years (24 to 25 EIA units, ns; 25 to 21, ns, respectively). At the age of 3 years, the children who turned seropositive for

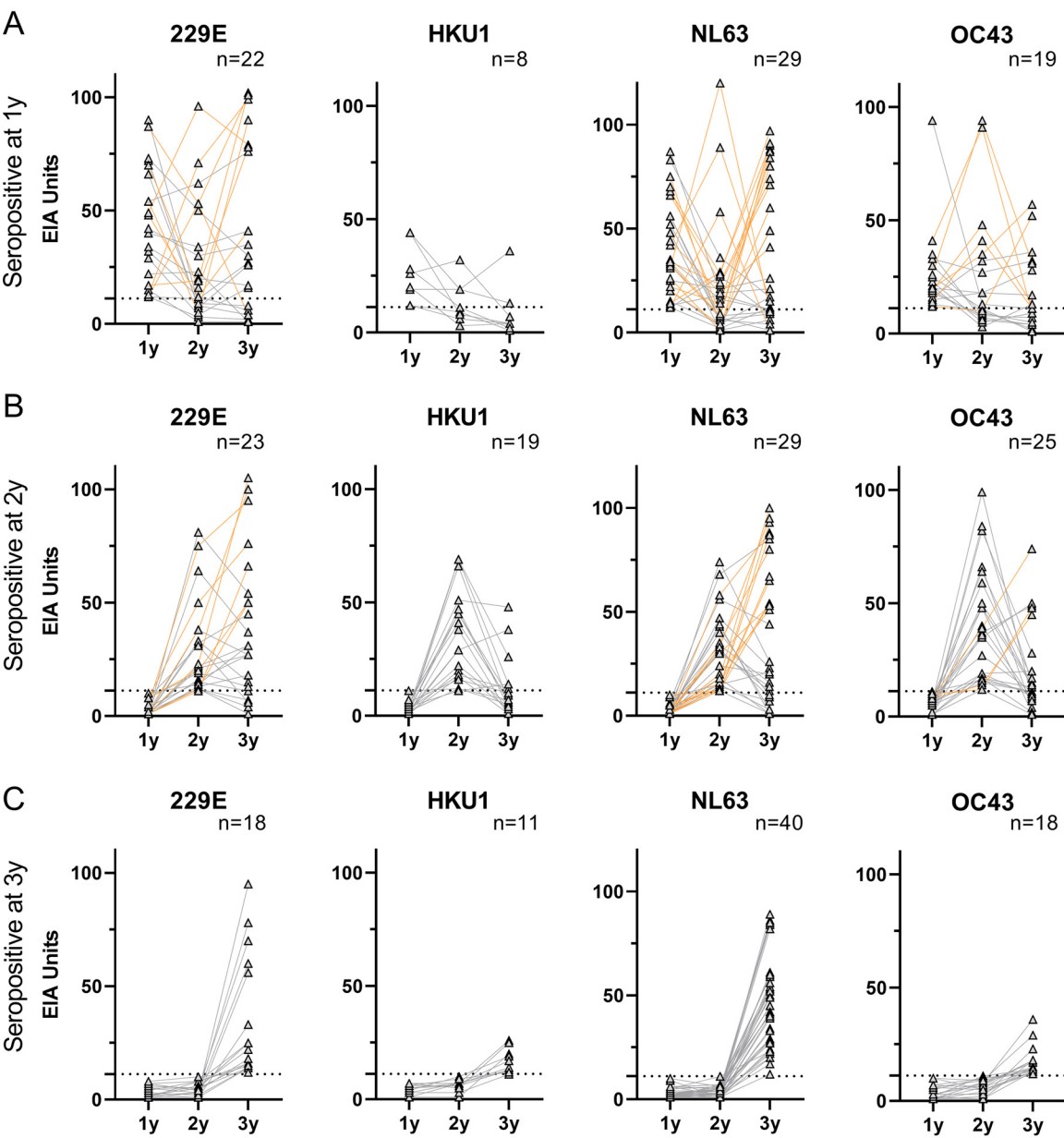

**FIG 2** IgG antibody levels in children following and preceding the primary infection. The children were separated into different groups based on the time they turned seropositive: before the age of 1 year (A), between 1 and 2 years (B), and between 2 and 3 years (C). The number of children in each group is presented in the graphs. Lines connect the IgG levels of an individual in each time point. Orange lines indicate individuals who have an increase of >20 EIA units in the antibody levels after initial seropositivity. Cutoff values are indicated as dashed lines. Statistical differences in IgG levels between different age groups were analyzed using Wilcoxon matched pairs signed-rank test. Two-tailed P-values *<0.05, **<0.01, ***<0.001, and ****<0.0001 were considered significant.

seasonal HCoVs between the age of 2 and 3 years, had higher GMALs than the children who had their primary infection earlier (Fig. 2A to C).

IgG antibody levels varied greatly between individuals after primary infection: some of the children turned clearly seronegative and others had significant increases in their antibody levels. This variation indicates either a decline in antibody levels in the absence of a reinfection or an increase in antibody levels due to a reinfection between sample collections.

Serum samples from children who were seropositive for a seasonal HCoV at 1 or 2 years were further analyzed for reinfection, which was defined as an increase of ≥20 EIA units in comparison to the previous serum sample. Significantly higher GMALs were observed in

children who had an indication of a reinfection although some groups of children were too small for statistical analysis (Fig. S1 in the supplemental material). The antibody levels of 1-year-old seropositive children increased following a reinfection between 1 and 2 years but decreased by the age of 3 years. In children who were seropositive for an HCoV by 2 years and showed no indication of a reinfection, the GMALs decreased to low levels (8-16 EIA units) by the age of 3 years (Fig. S1 and S2).

To analyze the kinetics of the decline of anti-HCoV IgG antibodies, the seropositive children with no indication of reinfection were divided into two groups based on the age they had turned seropositive. Following seropositivity at the age of 1 year, the GMALs for any HCoV decreased by 54–73% until the age of 2 years (35 to 12 EIA units for 229E, $P = 0.0017$; 23 to 11 for HKU1, ns; 32 to 9 for NL63, $P = 0.005$; 23 to 10 for OC43, $P = 0.011$, respectively; Fig. S2A in the supplemental material). The following year GMALs remained low in all of these groups. A similar decline in GMALs (70–77%) was seen from 2 to 3 years in children with a primary seropositivity at 2 years of age for HKU1 (29 to 7 EIA units, $P = 0.0003$), NL63 (31 to 9, $P = 0.0007$) and OC43 (34 to 10, $P = 0.0001$; Fig. S2B). For 229E, GMAL of children with a primary seropositivity at 2 years of age decreased by 31% (23 to 16 EIA units, ns), which is less than the decrease in antibody levels for other HCoVs. This reduced decrease in the GMAL was affected by several samples which, instead of a decrease, showed a weak (<20 EIA units) antibody increase (Fig. S1), potentially due to a reinfection that failed to increase IgG antibody levels over the reinfection cutoff or cross-reactive antibodies.

**Correlation of anti-N antibody levels between different virus types.** To evaluate potential cross-reactivity between anti-N protein antibodies, we calculated the correlation coefficients between EIA data for all serum samples ($n = 420$). The levels of IgG antibodies against 229E correlated moderately with the levels of anti-NL63 antibodies ($r = 0.4388$; Fig. 3). The correlation between the levels of anti-OC43 and anti-HKU1 antibodies was relatively strong ($r = 0.7003$), while the correlation rates of other anti-N protein antibody pairs were low ($r < 0.18$). If samples that were negative for both of the assays were removed from the calculation, the correlation between anti-OC43 and anti-HKU1 antibodies remained relatively strong (Fig. S3 in the supplemental material). However, the correlation between anti-229E and anti-NL63 antibodies weakened drastically, suggesting that negative samples may overemphasize the correlations between the assay pairs. The greater identities in N amino acid sequence of 229E and NL63 (48%) and of OC43 and HKU1 (66%) in comparison with their identity to other HCoVs (22-32%; Fig. S4) may contribute to better correlation coefficients.

**Validation of 229E and OC43 EIA with IFA.** To validate the EIA results we set up an IFA using 229E and OC43 virus-infected Huh7 cells (Fig. 4A). We analyzed 30 EIA-positive and 30 EIA-negative (randomly selected) serum samples to estimate the presence of whole-virus-specific antibodies and to determine IF antibody titer. The concordance between negative and positive samples in 229E IFA and EIA was 83% (20/30 of 229E EIA-positive and 0/30 of the 229E EIA-negative samples were positive in 229E IFA; Fig. 4B). Between OC43 IFA and EIA the concordance was also 83% (30/30 of OC43 EIA-positive and 10/30 OC43 EIA-negative samples were OC43 IFA-positive). Higher IFA titers were detected for EIA-positive samples in OC43 IFA than in 229E IFA (geometric mean titer 182 for OC43 and 37 for 229E) although in EIA the corresponding samples had lower geometric mean antibody levels for OC43 than for 229E (31 and 42 EIA units). The IFA titers for 229E and OC43 specific antibodies correlated well with the corresponding EIA unit values (229E EIA and IFA $r = 0.7258$, and OC43 EIA and IFA $r = 0.7809$; Fig. 4C).

**Early childhood seropositivity and reinfections for seasonal HCoVs.** IgG antibodies against 229E, HKU1, NL63 and OC43 N-proteins were detected in 16% (23/140), 6% (8/140), 21% (30/140) and 14% (20/140) of children by the age of 1 year (Fig. 5). During the follow up, the number of seropositive children increased: the ratio of children who turned seropositive between 1 and 2 years and 2 and 3 years were 16% and 13% for 229E, 14% and 8% for HKU1, 19% and 30% for NL63, and 18% and 12% for OC43, respectively.

Increases in antibody levels indicated that 36% (16/45) of the 229E-seropositive children had two or three 229E-infections by the age of 3 years. For HKU1, NL63 and OC43

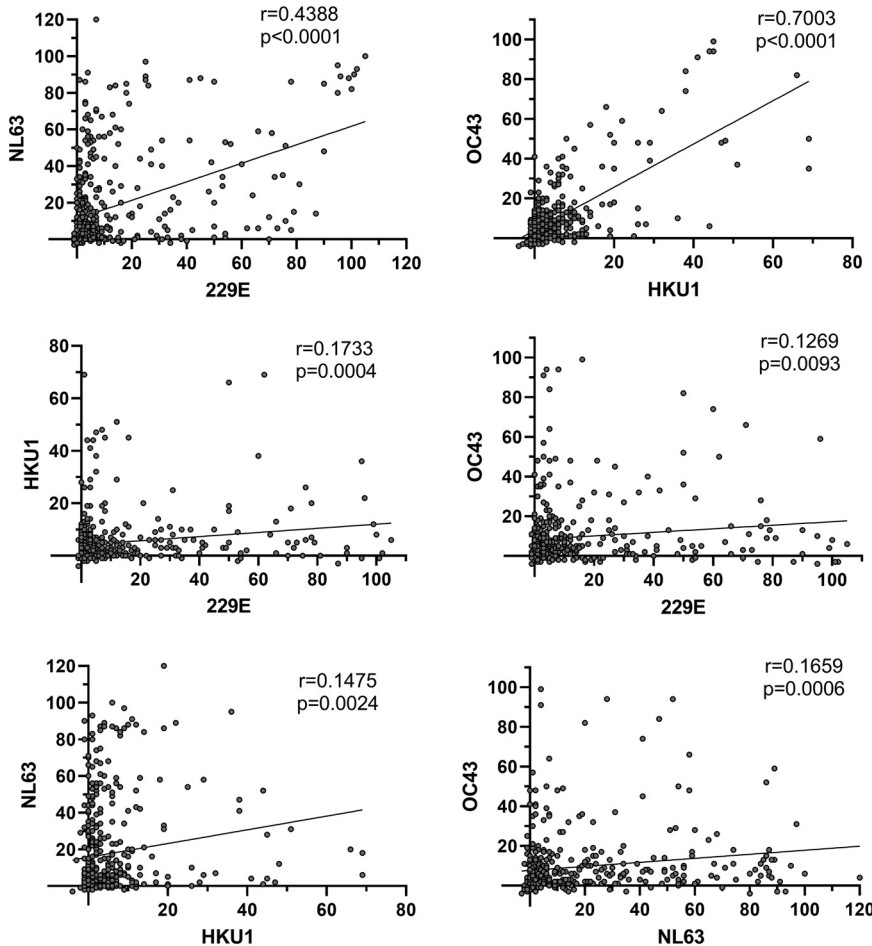

**FIG 3** Correlation of anti-HCoV N IgG antibody levels between different HCoV types. Correlation coefficients of IgG antibody levels for anti-HCoV N assays were determined with Pearson's correlation test. Calculations were done for all samples (*n* = 420) including negative EIA unit values that were given an arbitrary value of 1. Correlation coefficients (r), *P* values and linear regression lines are shown.

the corresponding reinfection rates were 0% (0/38), 48% (28/58) and 20% (9/44), respectively. One child showed an increase in 229E specific IgG antibody levels between 1 and 2 years of age as well as between 2 and 3 years of age indicating two reinfections. A summary of the numbers and percentages of infections and reinfections in different age groups is presented in Fig. 5.

## DISCUSSION

In this study, we estimated seasonal HCoV (229E, HKU1, OC43, and NL63) infection and reinfection rates by analyzing the presence of anti-coronavirus antibodies in serially collected serum specimens from 140 children between 1 and 3 years of age. Full-length N was selected as antigen for the EIA because it (i) is conserved within coronavirus species, (ii) is highly immunogenic and abundantly expressed, (iii) is relatively different from one coronavirus species to another, and (iv) includes all N epitopes. Previously, the C-terminal part of the N was used as a more specific antigen to avoid cross-reactivity between coronavirus species (5, 12). However, a recent work including study participants with sequential samples suggested that an assay with the C-terminal part of N as an antigen may also detect antibodies elicited against other seasonal coronavirus species (10), which promoted the selection of full-length N proteins as antigens for this study.

Previous studies have reported observations of cross-reactivity in serological responses to HCoVs (13, 14). The higher correlation coefficient rates between HKU1 and OC43 or 229E

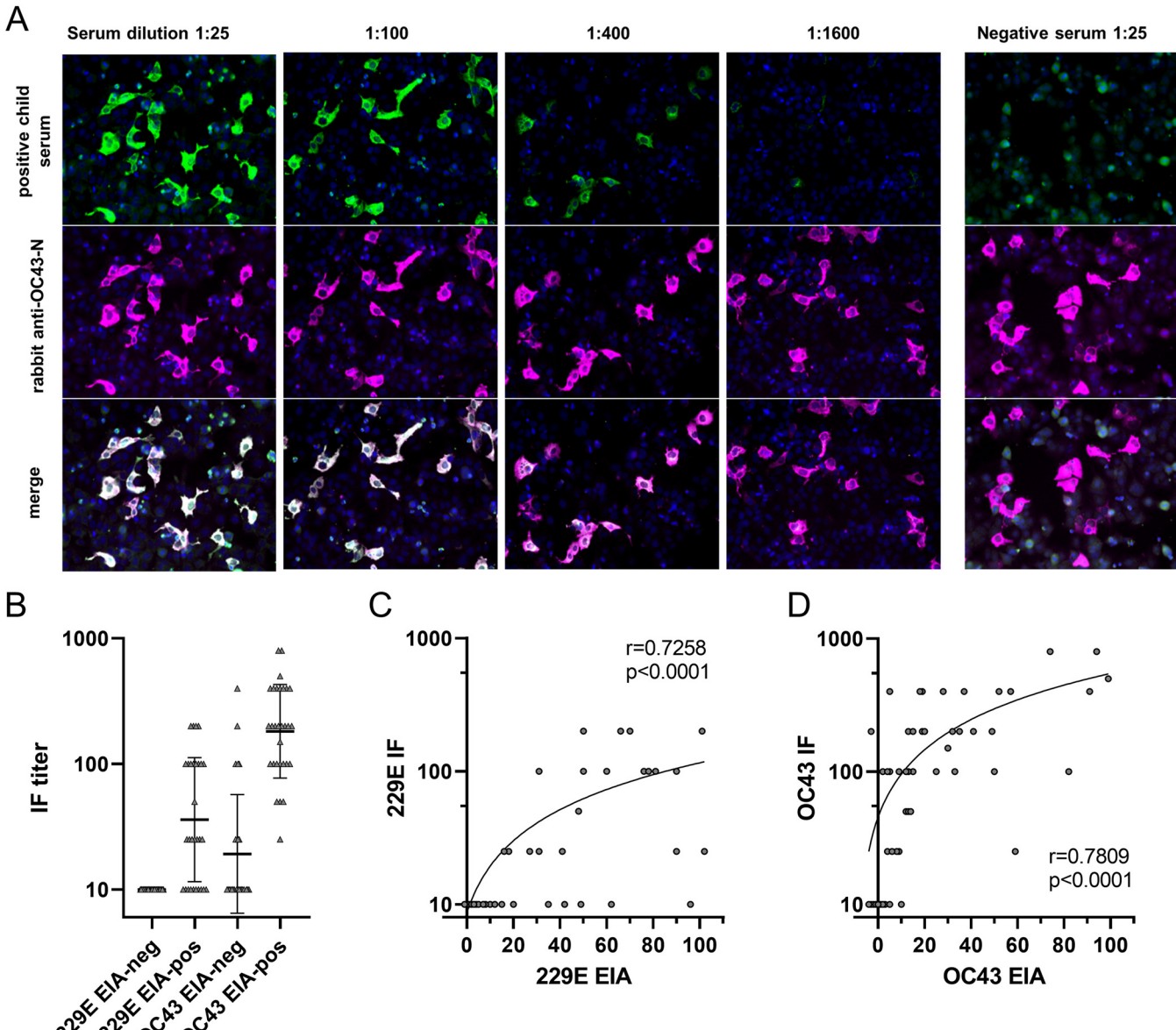

**FIG 4** An example of a dilution series of an IgG positive serum for OC43 and a negative serum sample (A). Permeabilized virus-infected cells were incubated with serum dilutions and with rabbit anti-OC43-N, followed by secondary anti-human (green) and anti-rabbit (magenta) antibodies. Cell nuclei were labeled with DAPI. IgG antibody titers of 60 serum samples detected with OC43 or 229E virus-infected Huh-7 cell line based IFA (B). Correlation between IgG levels in EIA and antibody titers in IFA for 229E-EIA and 229E-IFA as well as OC43-EIA and OC43-IFA was analyzed (C). Spearman's ranked correlation test coefficients (r), *P* values and regression lines are shown. (D) IgG antibodies detected for OC43 and 229E with indirect immunofluorescence assay (IFA).

and NL63 antibodies in our data indicated possible existence of cross-reactive antibodies recognizing N antigens of the same coronavirus genus. However, the knowledge on detailed infection history from each individual would be required to confirm the presence of cross-reactivity. To confirm the results from N-based EIA, we used IFA with virus-infected cells to detect 229E or OC43 virus-specific antibodies in the sera. The comparison of 229E IFA and EIA results indicates that for the detection of anti-229E IgG antibodies EIA is somewhat more sensitive than IFA. This observation was further apparent in the relatively high EIA unit levels in comparison to IFA titers. On the contrary, the IFA results with OC43 virus-infected cells suggested that OC43 anti-N protein IgG EIA may miss some OC43 antibody positive sera. It may be that 229E N is relatively more immunogenic in comparison to OC43 N protein or that the folding of the OC43 N protein in IFA versus EIA affects the binding of the antibodies. Despite a good correlation of EIA and IFA for 229E and OC43, the lack of IFA for HKU1 and especially for NL63 limits the IFA-based validation of EIA in this study. Unlike some recent

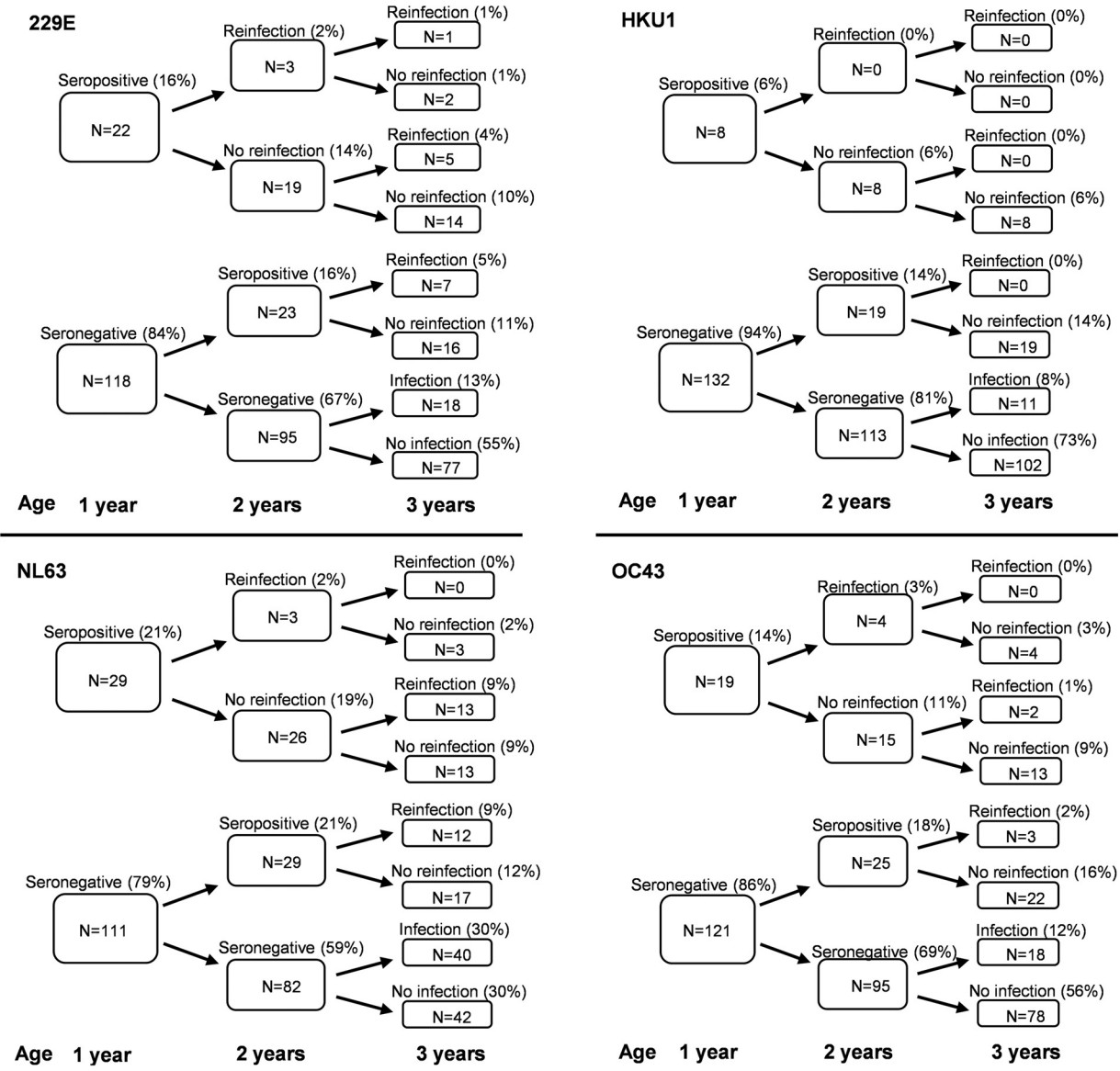

**FIG 5** Summary of seasonal coronavirus infections and reinfections in 140 children with follow-up samples at ages 1, 2 and 3 years. Status of seropositivity, infection, or reinfection for 229E, HKU1, NL63, and OC43 are based on anti-HCoV IgG antibody results in N antibody-specific enzyme immunoassay (EIA). Reinfection is determined as a significant (>20 EIA units) increase in serum anti-HCoV IgG antibody levels between the sequential serum samples. The numbers refer to the number of seropositive and reinfected children. The percentages refer to the percentage of children from the whole cohort (n = 140).

studies (15–17), we did not detect cross-reactivity between anti-SARS-CoV-2 and anti-seasonal HCoV N protein antibodies. This is likely due to the relatively stringent criteria used in this study for the calculation of the cutoff and in estimating reinfections to ensure the detection of only the coronavirus species-specific antibodies.

We observed that antibodies against seasonal HCoVs start to develop at an early age and the majority of children have antibodies against at least one HCoV by the age of 3 years. The participating children had no underlying diseases and thus this study provides information on the circulation of HCoVs in the general child population thereby avoiding biases caused by symptom-based sampling. Also, since the loss of maternal antibodies against seasonal HCoVs occurs by the age of 6 months (18, 19), the sera collected from children at 1 year of age (13 months) or later likely have antibodies that were developed following infections by circulating HCoVs.

By the age of 1 year, 6–21% of children were seropositive against different coronavirus types, which represent numbers lower than the ones reported in other age-stratified

studies (19, 20). It is possible that a 9-month maternity/parental leave and the relatively long home-care in Finland reduce the infection burden in very young children. However, the cumulative seroprevalence increased to 27–70% by the age of 3 years, which is more in line with previously reported data (5, 6, 20). In older children and adults the seroprevalence rates for 229E, NL63 and OC43 have been shown to reach near 100% (6, 17, 19, 20). In our study group the sampling was done between the years 2008 and 2014, and the observed HCoV infections, as well as reinfections, were most commonly caused by NL63 followed by almost as frequently with 229E and OC43. We observed a very low seroprevalence for HKU1, which is in line with some of the previous studies showing <25% seroprevalence for HKU1 in adults (13, 21) a figure that is quite different from the nearly 60% seropositivity rate reported by Zhou et al. in children aged 1 to 3 years (6). Geographic variation in circulating coronaviruses exists, and thereby it is possible that HKU1 is more prevalent in East Asian countries compared to Europe.

At least 36, 48, and 20% of the children with 229E, NL63 and OC43 infection had evidence of a reinfection by the same virus species during the following years. High frequency of reinfections may indicate that the humoral immune response to HCoV infections is relatively weak and primary infections do not build up an efficient protective immunity in children. In a previous study in serially sampled adults, reinfections were detected at the earliest 6 months after a previous infection, although the majority of reinfections occurred 12 months or later after the initial infection (10). A delay before the occurrence of a reinfection could suggest protection induced by the primary infection. Despite our observation of low rate of reinfections before the age of 2 years, more data is needed to evaluate the protective effect of the primary infection against a reinfection. Overall, following the primary infection, the antibody levels varied greatly between individuals and HCoV species. The GMALs were higher against alphacoronaviruses, which is likely due to a higher prevalence of these infections in the study population. Furthermore, the GMALs were significantly higher in children after a reinfection with any seasonal HCoV, which suggests that despite a decrease in the antibody levels after primary infection, the antibody response to a reinfection is robust.

Coronavirus infection-induced neutralizing antibodies have been shown to correlate with protection against a clinical infection (11), although no general level or cutoff for protection against reinfection has been defined. The immune system of children under 1 year of age is still developing and may not elicit a robust antibody response against viruses. In the majority of children, coronavirus infection-induced IgG antibody levels waned fairly rapidly. This observation is consistent with previous studies (10, 11) and waning immunity is likely a significant factor in explaining the frequency of reinfections. Interestingly, a comparable decline in IgG antibodies occurred in children independent of HCoV species or the child's age at the time of a primary infection, indicating that IgG antibody responses to the primary infection by 229E, HKU1, NL63 or OC43 are similar to each other in 1- to 2-year-old children.

In this work, we analyzed HCoV IgG antibody levels in children and demonstrated that they are highly susceptible to seasonal HCoV infections. In addition, reinfections are relatively common already in early life. We showed that the anti-HCoV IgG antibody levels decrease rapidly after a primary infection. Relatively rapid decay of infection-induced IgG antibodies and a high rate of reinfections suggests that antibody-mediated immunity may be essential in immune protection against HCoV infections, although also T cell-mediated immunity contributes to the immunity against infections. The short durability of anti-HCoV antibodies and susceptibility to reinfections may also apply to SARS-CoV-2 infection in children. If the duration of SARS-CoV-2 vaccine-induced response is comparable to the duration observed here with seasonal coronavirus infections, administration of booster vaccine doses will be needed to maintain protective levels of antibodies among vaccinated children.

## MATERIALS AND METHODS

**Study participants.** Study participants were children in the STEPS study (22), a prospective observational birth-cohort study in the Hospital District of Southwest Finland. One of the aims in the STEPS study is to assess the burden of infectious diseases in childhood. Children were born in 2008 to 2010

and they were recruited in the study before or soon after birth with no selection criteria other than the language (Finnish or Swedish) of the parents. A total of 420 serum samples from 140 children with sampling at 13, 24, and 36 months of age during visits to a study clinic were included into this study. The Ethics Committee of the Hospital District of Southwest Finland approved the study (decision no. 16/180/2008). Parents of participating children gave their written, informed consent. The sera have been analyzed previously for the seroprevalence of respiratory syncytial, influenza, and adenoviruses (23, 24).

**HCoV anti-N enzyme immunoassays.** Serum samples were tested for the presence of HCoV N specific IgG antibodies (anti-HCoV-N antibodies) by enzyme immunoassay (EIA) as described previously (25) with a slight modification: a 1:4000 dilution of horseradish peroxidase (HRP) conjugated anti-human antibodies was used. This adjustment to the protocol was done to optimize the assay for measurement of lower IgG concentrations in child sera in comparison to adult sera. Full-length N proteins, produced as fusion proteins with an N-terminal glutathione *S*-transferase protein (25), were used as the EIA antigens in this study.

The conversion of optical density (OD) values into EIA units was done by comparing the sample OD values to the OD values of positive (marked as 100) and negative (marked as 0) control sample pools measured by anti-229E-N EIA. Samples with an EIA unit values lower than 1 were marked as 1, in order to enable the calculation of geometric means. The cutoff value was determined as the mean of anti-SARS-CoV-2-N EIA value results from 13-month samples plus five standard deviations (SD). The interassay variation was <10% and thus, an increase of antibody level of ≥20 EIA units was considered to indicate a reinfection.

**229E and OC43 stock virus titration by focus-forming assay (FFA).** 229E (GenBank Accession OK662398) was propagated in human hepatoma cells (Huh7) and OC43 (GenBank Accession OK662397) in MRC-5 cells. The number of focus-forming units (FFU) was determined in the stock virus by a focus-forming assay (FFA). Briefly, black 96-well plates (Thermo Scientific) were seeded with Huh7 cells in infection medium consisting of d-MEM (Lonza) supplemented with 2% fetal calf serum (Gibco), 2 mM L-glutamine (Gibco) and penicillin/streptomycin (Gibco). The next day 10-fold dilution series of the virus stock was added to the cells. After 24 h of incubation, the cells were fixed with 4% paraformaldehyde and permeabilized with 0.1% Triton X-100 in PBS. A dilution of 1:1000 of polyclonal rabbit anti-229E-N or rabbit anti-OC43-N antibody (Huttunen et al., manuscript) was incubated for 30 min followed by washing of the cells for three times with 0.5% BSA in PBS. Secondary antibodies (1:1000 dilution of goat anti-rabbit IgG 564, Thermo Scientific) and DAPI (1:2500 dilution, Thermo Scientific) were added and incubated for 1h, and the wells were washed three times with 0.5% BSA in PBS. The number of FFUs was visualized and counted using EVOS FL Auto Fluorescence Inverted Microscope (Life Technologies).

**Indirect immunofluorescent assay (IFA) for OC43 and 229E.** Infection of Huh7 cells for IFA was done as in FFA (see above) using 0.2 FFU/cell of OC43 or 229E in infection medium. The cells were fixed and permeabilized as in FFA. Serum samples were diluted 1:25, 1:100, 1:400 and 1:1600 into PBS supplemented with 3% bovine serum albumin (BSA). Dilution series of a serum sample was added to cells in duplicates, and incubated for 30 min, washed and 1:1000 dilution of polyclonal rabbit 229E anti-N-GST or rabbit OC43 anti-N-GST antibodies (Huttunen et al., manuscript) were added for another 30 min. The detection of bound antibodies was done as in FFA using Alexa Fluor 488 conjugated goat anti-human IgG, Alexa Fluor 564 conjugated goat anti-rabbit IgG and DAPI (Thermo Scientific). Binding of serum IgG antibodies to virus antigens in infected cells was visualized using EVOS FL Auto Fluorescence Inverted Microscope (Life Technologies). Serum samples that were positive in IFA with a dilution of 1:25 or higher were considered IFA-positive. IgG antibody titer was defined as the highest dilution with visually detectable antibodies.

**Sequencing.** Virus stocks for 229E and OC43 were sequenced by the next-generation sequencing. The sequencing libraries were prepared using NEBNext Ultra II RNA Library Prep Kit for Illumina (New England Biolabs) following the manufacturer's instructions and sequenced with Illumina MiSeq sequencer using MiSeq reagent kit v3 with 250 bp reads. Adapter, low quality (quality score <30) and short (<50 nt) sequences were removed using Trimmomatic (26), followed by assembly using BWA-MEM (27), variant calling using LoFreq (28) and consensus calling using SAMtools (29) implemented in HaVoC pipeline (30). GenBank sequences NC_002645 and NC_006213 were used as reference sequences in the assembly for 229E and OC43, respectively. The virus stock sequence for 229E differed by 22 and for OC43 by 33 nucleotides in comparison to the GenBank reference sequence.

**Statistical analysis.** We used GraphPad Prism 8 software as the platform for statistical analyses of differences between the groups using Wilcoxon matched pairs signed-rank test corrected with Pratt's method and of correlations between the assays with Pearson's or Spearman's correlation test. Two-tailed *P*-values <0.05 were considered statistically significant.

**Data availability.** The source data used in this study are available from the corresponding author, PK, upon reasonable request. The GenBank accession codes for our 229E and OC43 viruses are OK662398 and OK662397, respectively.

## SUPPLEMENTAL MATERIAL

Supplemental material is available online only.
**SUPPLEMENTAL FILE 1**, PDF file, 1.1 MB.

## ACKNOWLEDGMENTS

Sari Maljanen is acknowledged for technical support. We thank all the children and their parents for participation in the STEPS study. This work was supported by the Jane and Aatos Erkko Foundation (grant numbers 3067-84b53 and 5360-cc2fc to I.J.), the

Sigrid Juselius Foundation (to I.J. and L.K.), and the Medical Research Council of the Academy of Finland (grant numbers 337530 and 336410 to I.J.).

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
