## [Reviewer comments · Microbiology Spectrum]

Microbiology Spectrum

Serological follow-up study indicates high seasonal coronavirus infection and reinfection rates in early childhood

Pekka Kolehmainen, Jemna Heroum, Pinja Jalkanen, Moona Huttunen, Laura Toivonen, Varpu Marjomäki, Matti Waris, Teemu Smura, Laura Kakkola, Sisko Tauriainen, Ville Peltola, and Ilkka Julkunen

Corresponding Author(s): Pekka Kolehmainen, University of Turku

Review Timeline:

Submission Date:	October 30, 2021
Editorial Decision:	November 27, 2021
Revision Received:	January 19, 2022
Editorial Decision:	February 4, 2022
Revision Received:	February 17, 2022
Accepted:	February 17, 2022

Editor: Takamasa Ueno

Reviewer(s): Disclosure of reviewer identity is with reference to reviewer comments included in decision letter(s). The following individuals involved in review of your submission have agreed to reveal their identity: Koji Watanabe (Reviewer #1)

Transaction Report:

DOI: <https://doi.org/10.1128/spectrum.01967-21>

November 27, 2021

Dr. Pekka Kolehmainen
University of Turku
Institute of Biomedicine
Kiinamylynkatu 10
Turku
Finland

Re: Spectrum01967-21 (Serological follow-up study indicates high seasonal coronavirus infection and reinfection rates in early childhood)

Dear Dr. Pekka Kolehmainen:

Link Not Available

Sincerely,

Takamasa Ueno

Journals Department
Reviewer comments:

Reviewer #1 (Comments for the Author):

Kolehmainen P, et al. prospectively collected serum samples from 140 children in their birth cohort of STEP study, and evaluated antibody titers for 4 different genotypes of seasonal human coronaviruses (HCoVs). They showed the high prevalence of HCoVs in their cohort, especially NL63 is highly prevalent in the young age (< 3 y.o.). These epidemiological data itself has much importance in this research field. However, most of the results and discussions should be reconstructed before considering the formal review.

Major comments:

1. What is the importance of "geometric mean IgG" in results section (line 158-182) and figure 1A in this paper? All geometric mean IgG titer other than year 3 of NL63 are much lower than cut-off titer. Also, are the differences between different years statistically significant? If yes, please indicate (e.g. *) in the figure 1A. Also, please specify the reason why you presented these data.

2. Please add the seropositivity of each year (bar) and cumulative seropositivity (triangle) for "any of 4 genotypes" in figure 1B. I, as a representative of the reader, want to know how often any of HCoVs infection occurs in this population.
3. I feel that figure 3 (possibly figure 2) could be deleted, because figure 2 & 3A only show just the parts of figure 1. Also, we can know the frequency of reinfection from figure 6. Moreover, we cannot know the transitional changes of antibody titer of each individual even from figure 2 & 3A. Dots plot without transitional changes (e.g. line between years) for individuals and the titers of geometric mean IgG do not seem very important in this paper.
4. Results section is too long. The authors just checked HCoVs IgG titer for 3 years, and checked their correlation with IFA assay. Epidemiology from the serological data of N-protein specific IgG antibody is the main data, which is well-summarized in the last paragraph (line 288-300) & figure 6. On the other hand, I couldn't find any importance from the detailed description about primary infection and re-infection in the 2nd and 3rd paragraphs (line 184-240).
5. In the 6th paragraph of the results section (line 274-286), the author should clarify the positive/negative match rates between IFA and EIA in each genotype. It is extremely difficult to validate EIA from figure 5B and C. Correlation between 2 methods, and effect of EIA titer on them should be discussed after showing positive/negative match rates. Also, validation of EIA to IFA was not assessed about NL63 (and HKU1), which is the most prevalent genotype in this area. Please mention the fact as limitation in discussion section.

Minor comments:

1. If you compare the value, please specify whether it is statistically significant, or not. For example, mean geometric antibody titers (line 166, 169-175, 226...)
2. Please specify the definition of IFA positive. 1:25 positive sample is positive for IFA, or is there another cut-off?
3. In this study, clinical outcome (symptoms, lab data, and chest Xp, etc.) was not assessed. It is never known whether IgG elevation (and re-elevation) occurred asymptotically, or symptomatically. I wonder the definition the "protection" and "re-infection" in this study. They discussed rapid decline of the antibody titer can contribute to the sensitivity to the same virus, resulting in the high rates of re-infection (line 353-370). However, you should not know the study participants have any symptoms at the re-elevation of antibody titer. It is definitely true re-elevation of antibody titer is caused by the re-exposure to the pathogen, however, we are not able to say re-infection due to the lost of protection (the rapid decline of the antibody titer after primary infection). These descriptions should be deleted, or rewrite with "re-exposure", not "re-infection".

Reviewer #2 (Comments for the Author):

1. Lines 60-61: Should be rephrased, including other parts that have language issues.
2. Lines 160-167: among the four HCoVs 229E, HKU1, NL63, and OC43, indicate in the text which was most commonly detected among the children at year 1, 2 and 3. It will be informative for the authors to show the frequency of HCoVs at each time point. Also, the authors should at least state the co-infections patterns i.e. which pairs of HCoVs predominate within this population at each time point and overall.
3. Lines 179-180: what is the proportion of children who tested seronegative after initial seropositivity? The authors should clearly indicate specific HCoVs that showed this trend and their proportions.
4. What are the possible reasons for the differential patterns of the mean IgG antibody levels of the different HCoVs versus age during primary infection?
5. Lines 266: What is the clinical significance of the correlation between anti-OC43 and anti-HKU1 antibodies.
6. From the correlation of 229E and NL63, and OC43 and HKU1 EIA how can you establish the potential cross-reactivity phenomenon?
7. Since reinfection for most HCoVs was observed by year 3, how can the authors distinguish the role of persistence of humoral immunity versus risk exposure?

Staff Comments:

Preparing Revision Guidelines

Please return the manuscript within 60 days; if you cannot complete the modification within this time period, please contact me. If you do not wish to modify the manuscript and prefer to submit it to another journal, please notify me of your decision immediately so that the manuscript may be formally withdrawn from consideration by Microbiology Spectrum.

Kolehmainen P, et al. prospectively collected serum samples from 140 children in their birth cohort of STEP study, and evaluated antibody titers for 4 different genotypes of seasonal human coronaviruses (HCoVs). They showed the high prevalence of HCoVs in their cohort, especially NL63 is highly prevalent in the young age (< 3 y.o.). These epidemiological data itself has much importance in this research field. However, most of the results and discussions should be reconstructed before considering the formal review.

Major comments:

1. What is the importance of “geometric mean IgG” in results section (line 158–182) and figure 1A in this paper? All geometric mean IgG titer other than year 3 of NL63 are much lower than cut-off titer. Also, are the differences between different years statistically significant? If yes, please indicate (e.g. *) in the figure 1A. Also, please specify the reason why you presented these data.
2. Please add the seropositivity of each year (bar) and cumulative seropositivity (triangle) for “any of 4 genotypes” in figure 1B. I, as a representative of the reader, want to know how often any of HCoVs infection occurs in this population.
3. I feel that figure 3 (possibly figure 2) could be deleted, because figure 2 & 3A only show just the parts of figure 1. Also, we can know the frequency of reinfection from figure 6. Moreover, we cannot know the transitional changes of antibody titer of each individual even from figure 2 & 3A. Dots plot without transitional changes (e.g. line between years) for individuals and the titers of geometric mean IgG do not seem very important in this paper.
4. Results section is too long. The authors just checked HCoVs IgG titer for 3 years, and checked their correlation with IFA assay. Epidemiology from the serological data of N-protein specific IgG antibody is the main data, which is well-summarized in the last paragraph (line 288–300) & figure 6. On the other hand, I couldn't find any importance from the detailed description about primary infection and re-infection in the 2nd and 3rd paragraphs (line 184–240).
5. In the 6th paragraph of the results section (line 274–286), the author should clarify the positive/negative match rates between IFA and EIA in each genotype. It is extremely difficult to validate EIA from figure 5B and C. Correlation between 2 methods, and effect of EIA titer on them should be discussed after showing positive/negative match rates. Also, validation of EIA to IFA was not assessed about NL63 (and HKU1), which is the most prevalent genotype in this area. Please mention the fact as limitation in discussion section.

Minor comments:

1. If you compare the value, please specify whether it is statistically significant, or not. For example, mean geometric antibody titers (line 166, 169–175, 226...)
2. Please specify the definition of IFA positive. 1:25 positive sample is positive for IFA, or is there another cut-off?
3. In this study, clinical outcome (symptoms, lab data, and chest Xp, etc.) was not assessed. It is never known whether IgG elevation (and re-elevation) occurred asymptotically, or symptomatically. I wonder the definition the “protection” and “re-infection” in this study. They discussed rapid decline of the antibody titer can contribute to the sensitivity to the same virus, resulting in the high rates of re-infection (line 353–370). However, you should not know the study participants have any symptoms at the re-elevation of antibody titer. It is definitely true re-elevation of antibody titer is caused by the re-exposure to the pathogen, however, we are not able to say re-infection due to the lost of protection (the rapid decline of the antibody titer after primary infection). These descriptions should be deleted, or rewrite with “re-exposure”, not “re-infection” .

Dr. Takamasa Ueno

Editor

Microbiology Spectrum

Re: Spectrum01967-21 "Serological follow-up study indicates high seasonal coronavirus infection and reinfection rates in early childhood"

We thank the Editor and the reviewers for their careful evaluation of our manuscript. We have now modified the manuscript according to the reviewers' suggestions and think the changes have greatly improved the manuscript. Modifications in the revised manuscript are highlighted.

We have responded to the reviewers' comments as detailed below:

Reviewer #1 (Comments for the Author):

Kolehmainen P, et al. prospectively collected serum samples from 140 children in their birth cohort of STEP study, and evaluated antibody titers for 4 different genotypes of seasonal human coronaviruses (HCoVs). They showed the high prevalence of HCoVs in their cohort, especially NL63 is highly prevalent in the young age (< 3 y.o.). These epidemiological data itself has much importance in this research field. However, most of the results and discussions should be reconstructed before considering the formal review.

We thank the reviewer #1 for constructive comments and his/her scientific view of the importance of this study for providing new data on the epidemiology of HCoVs. We modified the manuscript, concentrating mainly in the results section as suggested in the detailed comments by reviewer #1.

Major comments:

*1. What is the importance of "geometric mean IgG" in results section (line 158-182) and figure 1A in this paper? All geometric mean IgG titer other than year 3 of NL63 are much lower than cut-off titer. Also, are the differences between different years statistically significant? If yes, please indicate (e.g. *) in the figure 1A. Also, please specify the reason why you presented these data.*

We agree with the R#1 here that the geometric mean IgG values are not necessary in this section and therefore we modified the results section and figure 1A accordingly. The reason for the low GMT value is the fact that a majority of samples are negative and thus they remains below the cut-off. Data in 1A describes the big picture of HCoV antibody levels in 1-3 years aged children, including the clear increases in antibody levels as well as the lack of these increases in antibodies for MERS and SARS-CoV-2 (except for very few cases likely due to some cross-reactivity).

2. Please add the seropositivity of each year (bar) and cumulative seropositivity (triangle) for "any of 4 genotypes" in figure 1B. I, as a representative of the reader, want to know how often any of HCoV's infection occurs in this population.

Cumulative seropositivity data has now been added into figure 1B as suggested.

3. I feel that figure 3 (possibly figure 2) could be deleted, because figure 2 & 3A only show just the parts of figure 1. Also, we can know the frequency of reinfection from figure 6. Moreover, we cannot know the transitional changes of antibody titer of each individual even from figure 2 & 3A. Dots plot without transitional changes (e.g. line between years) for individuals and the titers of geometric mean IgG do not seem very important in this paper.

Our analysis for seropositivity as well as reinfection induced and decline in antibody levels is completely based on the IgG antibody levels and, therefore, we think that detailed presentation of these data strengthens the study. The key message in figure 2 is to show that a primary infection with any seasonal HCoV increases the antibody levels, which then start to decline as an average. Figure 2 also shows that there is great variation between individuals. We modified the text to concentrate more on the key message. It is possible to add lines to present transitional changes for an individual but the addition of lines makes figure 2 very busy and difficult to read (an example figure is shown below).

Our analysis on reinfections is based on the changes in antibody levels (>20 EIA units) after primary seropositivity and figure 3 shows these data in detail. Since the number of reinfections is the key message in this part of the text, we have moved the figure 3 graph set into supplementary figure 1. We feel that the data is relevant and convincing, but for the clarity of main text and message the data is presented in supplementary data.

4. Results section is too long. The authors just checked HCoV's IgG titer for 3 years, and checked their correlation with IFA assay. Epidemiology from the serological data of N-protein specific IgG antibody is the main data, which is well-summarized in the last

paragraph (line 288-300) & figure 6. On the other hand, I couldn't find any importance from the detailed description about primary infection and re-infection in the 2nd and 3rd paragraphs (line 184-240).

Our prospective cohort and study material allowed detailed follow-up of antibody changes in the study population, since serum specimens collected at ages 1, 2 and 3 years were included for all participants. This is a clear strength of our study and, therefore, we aimed to analyse all the obtained data in greater detail. We agree with the reviewer that the description of the findings could be more concise and thus we modified and shortened the text to describe only the main findings regarding changes in antibody levels. In the results section, we also combined all of the parts that describe antibody levels. Figure 2 shows that antibody levels may develop in different directions after the primary infection. This forms also the basis for observation of decline in the antibody levels as well as the reinfections. Furthermore the antibody levels develop differently between different HCoV, likely due to the different prevalence of infections/reinfections. We feel that detailed description of seasonal coronavirus antibody decline is very relevant, since similar questions will likely be raised in terms of childhood COVID-19. Our analyses can in the future be compared with immunity to COVID-19 in young children.

5. In the 6th paragraph of the results section (line 274-286), the author should clarify the positive/negative match rates between IFA and EIA in each genotype. It is extremely difficult to validate EIA from figure 5B and C. Correlation between 2 methods, and effect of EIA titer on them should be discussed after showing positive/negative match rates. Also, validation of EIA to IFA was not assessed about NL63 (and HKU1), which is the most prevalent genotype in this area. Please mention the fact as limitation in discussion section.

The results section describing comparison of IFA and EIA was modified as suggested. This type of validation was done in order to have a general view of the correlation of different methods. EIA is measuring antibodies against the used antigen (N protein) while IFA is measuring total immunity against all coronavirus proteins that are expressed in virus infected cells. The correlation between the methods was relatively good. Text regarding the lack of IFA for NL63 and HKU1 was included in the discussion.

Minor comments:

1. If you compare the value, please specify whether it is statistically significant, or not. For example, mean geometric antibody titers (line 166, 169-175, 226...)

Indication of statistical significance was included in parts of the text that compare values.

2. Please specify the definition of IFA positive. 1:25 positive sample is positive for IFA, or is there another cut-off?

We considered immunofluorescence positivity in 1:25 serum dilution to indicate seropositive in IFA. There is no generally defined titer for seropositivity for seasonal coronavirus abs

especially in young children. A sentence clarifying the definition of IFA-positivity was added into the manuscript.

3. In this study, clinical outcome (symptoms, lab data, and chest Xp, etc.) was not assessed. It is never known whether IgG elevation (and re-elevation) occurred asymptotically, or symptomatically. I wonder the definition the "protection" and "re-infection" in this study. They discussed rapid decline of the antibody titer can contribute to the sensitivity to the same virus, resulting in the high rates of re-infection (line 353-370). However, you should not know the study participants have any symptoms at the re-elevation of antibody titer. It is definitely true re-elevation of antibody titer is caused by the re-exposure to the pathogen, however, we are not able to say re-infection due to the lost of protection (the rapid decline of the antibody titer after primary infection). These descriptions should be deleted, or rewrite with "re-exposure", not "re-infection".

The term re-infection was chosen instead of re-exposure since we think that infection is needed for an activation of an immune response that results into increased levels of antibodies. We think that a re-exposure is an event where infection is directly blocked and no infection or replication of the virus is taking place in the upper respiratory tract. Many respiratory infections may be subclinical (silent or asymptomatic), but we think there is still a question of an infection. COVID-19 has in many instances been subclinical, but we think that we still describe the person as being infected. Similarly HCV infection is very often silent but we talk about cronicly infected individuals. We fully agree with the reviewer that there is a difference between symptomatic infection and asymptomatic re-exposure or re-infection. We do, however, prefer to use the term re-infection since the virus has to replicate in the host in order to induce good antibody response. We hope that the reviewer can accept this interpretation.

Reviewer #2 (Comments for the Author):

We are grateful for the Rewiever #2 for the detailed comments, which definitely enabled us to improve the manuscript.

1. Lines 60-61: Should be rephrased, including other parts that have language issues.

The sentence was rewritten and the language in the manuscript has now been improved.

2. Lines 160-167: among the four HCoV's 229E, HKU1, NL63, and OC43, indicate in the text which was most commonly detected among the children at year 1, 2 and 3. It will be informative for the authors to show the frequency of HCoV's at each time point. Also, the authors should at least state the co-infections patterns i.e. which pairs of HCoV's predominate within this population at each time point and overall.

The text in this paragraph was rearranged and modified to describe the prevalence of different HCoVVs instead of the differences in the geometric means in different age groups. Our analysis is a seroprevalence study for serum specimens collected at specific time points after birth. This approach does not give us an opportunity to evaluate the frequency of coinfections. We would like to point out that any of the children had not been admitted to hospital due to a coronavirus infection and thus most infections have likely been very mild or even asymptomatic.

3. Lines 179-180: what is the proportion of children who tested seronegative after initial seropositivity? The authors should clearly indicate specific HCoVVs that showed this trend and their proportions.

Data on proportions of children who tested seronegative after previous seropositivity has now been added into the text.

4. What are the possible reasons for the differential patterns of the mean IgG antibody levels of the different HCoVVs versus age during primary infection?

Discussion of possible reasons on antibody levels after primary infection in different ages has now been added into the text.

5. Lines 266: What is the clinical significance of the correlation between anti-OC43 and anti-HKU1 antibodies.

Analysis of antibody correlations was done because OC43 and HKU1 coronaviruses show higher sequence identity in their N protein amino acid sequence. Therefore, it is likely that there may be some cross-reactivity of antibodies between OC43 and HKU1 anti-N protein antibodies. HKU1 is very rare in Finland (and Europe) and thus HKU1 seropositivity likely is at least partially due to cross-reactivity between the strains. Analysis of antibody correlations gives us some information to evaluate HKU1 infection rates (which was likely very low).

6. From the correlation of 229E and NL63, and OC43 and HKU1 EIA how can you establish the potential cross-reactivity phenomenon?

Since we have no complete knowledge on the infection history from these individuals, our discussion about cross-reactivity is speculative. The cross-reactivity between seasonal HCoVVs has been shown before and it would support our observations regarding virus type-specific antibody correlations. We have modified the discussion section to state clearly that the presence of cross-reactivity was not (or could not be) confirmed here.

7. Since reinfection for most HCoV was observed by year 3, how can the authors distinguish the role of persistence of humoral immunity versus risk exposure?

To confirm that normal fluctuation in HCoV antibody levels could not be considered falsely as a reinfection we set the limit for reinfection relatively high (>20 EIA units). It is likely that we missed some reinfections due to the high cut-off but we chose this option to ensure that we observe only the clear cases of reinfection. Fluctuation in antibody levels in individuals who remained negative for a HCoV throughout the study period was limited to few EIA units. Additionally, in initially seropositive individuals (at 1 or 2 years of age) the antibody levels tended to either decrease or show a clear increase. In some individuals the antibody levels remained relatively constant or even weakly increased between the yearly samples, but we wanted to use relative stringent criteria to define reinfection. A similar approach was also adopted in our previous paper on RSV seroepidemiology (Kutsaya et al. *Epid. Infect.* 2017).

February 4, 2022

Dr. Pekka Kolehmainen
University of Turku
Institute of Biomedicine
Kiinamylynkatu 10
Turku
Finland

Re: Spectrum01967-21R1 (Serological follow-up study indicates high seasonal coronavirus infection and reinfection rates in early childhood)

Dear Dr. Pekka Kolehmainen:

Reviewer #1 still has some critical concerns in the revised form, especially in relation to interpretation to Figure 2. I would suggest to ask a statistic expert if necessary.

Link Not Available

Sincerely,

Takamasa Ueno

Journals Department
Reviewer comments:

Reviewer #1 (Comments for the Author):

Revised manuscript is well modified, and key messages were much clearer than the initially submitted manuscript. However, I still have some additional concerns especially about the figure presentation and its interpretation.

1. Figure 2A-C: Lines between different years should be presented in order to clarify the transitional changes of antibody titers in

each individual, because the main point of this figure is showing the changes in antibody titers at re-infection. I agree with your response on rebuttal letter that Figure 2 shows that antibody levels may develop in different directions after the primary infection. So "direction=lines between different years" should be presented. If you feel too busy for the figure, you can change the line colors (e.g. red) which represent re-infection. In that sense above, comparison of GMAL between different years (non-paired comparison of antibody titer) is nonmeaningful in this figure, because these data contain only some parts of the study population (people infected with HCoVs at 1st, 2nd, or 3rd year).

2. The reviewer could not understand the descriptions between line 335 and 339. If you see the protective effect of primary infection, you have to compare the infection rates at 2nd year between seropositive participants and seronegative ones at 1st year (e.g. $3/22=13.6\%$ and $23/118=19.5\%$ for 229E virus).

Also, if you see the duration of the protective effect of primary infection, you might see when re-infection occur after primary infection (e.g. $(3+7)/(22+23)=22.2\%$ within 1 year after primary infection, and $5/22=22.7\%$ between 1 and 2 years after primary infection, for 229E virus). It doesn't seem protective effect of primary infection is stronger for the first year.

3. Author said the GMALs are higher after re-infection than those after primary infection, which suggests.... (line 340-345) It seems quite natural because re-infection was defined as an increase of > 20 EIA units from previous year, and the author see the changes of antibody titers only 3 times in this study.

Staff Comments:

Preparing Revision Guidelines

Please return the manuscript within 60 days; if you cannot complete the modification within this time period, please contact me. If you do not wish to modify the manuscript and prefer to submit it to another journal, please notify me of your decision immediately so that the manuscript may be formally withdrawn from consideration by Microbiology Spectrum.

Revised manuscript is well modified, and key messages were much clearer than the initially submitted manuscript. However, I still have some additional concerns especially about the figure presentation and its interpretation.

1. Figure 2A-C: Lines between different years should be presented in order to clarify the transitional changes of antibody titers in each individual, because the main point of this figure is showing the changes in antibody titers at re-infection. I agree with your response on rebuttal letter that Figure 2 shows that antibody levels may develop in different directions after the primary infection. So “direction=lines between different years” should be presented. If you feel too busy for the figure, you can change the line colors (e.g. red) which represent re-infection. In that sense above, comparison of GMAL between different years (non-paired comparison of antibody titer) is nonmeaningful in this figure, because these data contain only some parts of the study population (people infected with HCoVs at 1st, 2nd, or 3rd year).
2. The reviewer could not understand the descriptions between line 335 and 339. If you see the protective effect of primary infection, you have to compare the infection rates at 2nd year between seropositive participants and seronegative ones at 1st year (e.g. $3/22=13.6\%$ and $23/118=19.5\%$ for 229E virus). Also, if you see the duration of the protective effect of primary infection, you might see when re-infection occur after primary infection (e.g. $(3+7)/(22+23)=22.2\%$ within 1 year after primary infection, and $5/22=22.7\%$ between 1 and 2 years after primary infection, for 229E virus). It doesn't seem protective effect of primary infection is stronger for the first year.
3. Author said the GMALs are higher after re-infection than those after primary infection, which suggests... (line 340-345) It seems quite natural because re-infection was defined as an increase of > 20 EIA units from previous year, and the author see the changes of antibody titers only 3 times in this study.

Dr. Takamasa Ueno

Editor

Microbiology Spectrum

Re: Spectrum01967-21R1 "Serological follow-up study indicates high seasonal coronavirus infection and reinfection rates in early childhood"

We thank the Editor and the reviewers for their additional evaluation of our manuscript. We have now further modified the manuscript according to the reviewers' suggestions and think the changes have greatly improved the manuscript. Modifications in the revised manuscript are highlighted.

We have responded to the reviewers' comments as detailed below:

Reviewer #1 (Comments for the Author):

Revised manuscript is well modified, and key messages were much clearer than the initially submitted manuscript. However, I still have some additional concerns especially about the figure presentation and its interpretation.

1. Figure 2A-C: Lines between different years should be presented in order to clarify the transitional changes of antibody titers in each individual, because the main point of this figure is showing the changes in antibody titers at re-infection. I agree with your response on rebuttal letter that Figure 2 shows that antibody levels may develop in different directions after the primary infection. So "direction=lines between different years" should be presented. If you feel too busy for the figure, you can change the line colors (e.g. red) which represent re-infection. In that sense above, comparison of GMAL between different years (non-paired comparison of antibody titer) is nonmeaningful in this figure, because these data contain only some parts of the study population (people infected with HCoV at 1st, 2nd, or 3rd year).

We modified the figures 2A-C as suggested by the reviewer. Connecting lines were added in the figure and lines for GMALs were removed. The analysis of significance between GMALs in different years was based on paired comparison of antibody levels and thus we suggest that the analysis of the data in figure 2 will be remaining. However, the addition of lines clarifies the changes in the antibody levels of a given individual through the study period. Figure legend was updated accordingly.

2. The reviewer could not understand the descriptions between line 335 and 339. If you see the protective effect of primary infection, you have to compare the infection rates at 2nd year between seropositive participants and seronegative ones at 1st year (e.g. $3/22=13.6\%$ and $23/118=19.5\%$ for 229E virus).

Also, if you see the duration of the protective effect of primary infection, you might see when re-infection occur after primary infection (e.g. $(3+7)/(22+23)=22.2\%$ within 1 year after primary infection, and $5/22=22.7\%$ between 1 and 2 years after primary infection, for 229E virus). It doesn't seem protective effect of primary infection is stronger for the first year.

We agree with the reviewer that no conclusion regarding protective effect of primary infection can be made from our data; however, larger data could enable drawing a stronger conclusion in either direction regarding this. We modified this section of the manuscript to include only discussion of the potential protective effect of primary infection.

3. Author said the GMALs are higher after re-infection than those after primary infection, which suggests.... (line 340-345) It seems quite natural because re-infection was defined as an increase of > 20 EIA units from previous year, and the author see the changes of antibody titers only 3 times in this study.

As the reviewer points out, this observation is clearly, what is expected due to the criteria set for a reinfection. However, for a child who turned seropositive by 1 year of age and experienced a reinfection between 2-3 years, an increase of > 20 EIA between 2-3 years does not automatically result in higher antibody level at 3 years in comparison to that at 1 year. Therefore, we think that it is somewhat relevant to mention this observation of higher antibody levels.

February 17, 2022

Dr. Pekka Kolehmainen
University of Turku
Institute of Biomedicine
Kiinamylynkatu 10
Turku
Finland

Re: Spectrum01967-21R2 (Serological follow-up study indicates high seasonal coronavirus infection and reinfection rates in early childhood)

Dear Dr. Pekka Kolehmainen:

Your manuscript has been accepted, and I am forwarding it to the ASM Journals Department for publication. You will be notified when your proofs are ready to be viewed.

Sincerely,

Takamasa Ueno
Editor, Microbiology Spectrum

Journals Department
Supplemental Material 1: Accept